# Crosstalk of Inflammatory Cytokines within the Breast Tumor Microenvironment

**DOI:** 10.3390/ijms24044002

**Published:** 2023-02-16

**Authors:** Ola Habanjar, Rea Bingula, Caroline Decombat, Mona Diab-Assaf, Florence Caldefie-Chezet, Laetitia Delort

**Affiliations:** 1Université Clermont-Auvergne, INRAE, UNH, Unité de Nutrition Humaine, CRNH-Auvergne, 63000 Clermont-Ferrand, France; 2Equipe Tumorigénèse Pharmacologie Moléculaire et Anticancéreuse, Faculté des Sciences II, Université Libanaise Fanar, Beyrouth 1500, Lebanon

**Keywords:** immune-inflammatory cells, adaptive immune cells, immunocompetent cells, cytokines, adipokines, interleukins, crosstalk, signaling pathways, tumor microenvironment

## Abstract

Several immune and immunocompetent cells, including dendritic cells, macrophages, adipocytes, natural killer cells, T cells, and B cells, are significantly correlated with the complex discipline of oncology. Cytotoxic innate and adaptive immune cells can block tumor proliferation, and others can prevent the immune system from rejecting malignant cells and provide a favorable environment for tumor progression. These cells communicate with the microenvironment through cytokines, a chemical messenger, in an endocrine, paracrine, or autocrine manner. These cytokines play an important role in health and disease, particularly in host immune responses to infection and inflammation. They include chemokines, interleukins (ILs), adipokines, interferons, colony-stimulating factors (CSFs), and tumor necrosis factor (TNF), which are produced by a wide range of cells, including immune cells, such as macrophages, B-cells, T-cells, and mast cells, as well as endothelial cells, fibroblasts, a variety of stromal cells, and some cancer cells. Cytokines play a crucial role in cancer and cancer-related inflammation, with direct and indirect effects on tumor antagonistic or tumor promoting functions. They have been extensively researched as immunostimulatory mediators to promote the generation, migration and recruitment of immune cells that contribute to an effective antitumor immune response or pro-tumor microenvironment. Thus, in many cancers such as breast cancer, cytokines including leptin, IL-1B, IL-6, IL-8, IL-23, IL-17, and IL-10 stimulate while others including IL-2, IL-12, and IFN-γ, inhibit cancer proliferation and/or invasion and enhance the body’s anti-tumor defense. Indeed, the multifactorial functions of cytokines in tumorigenesis will advance our understanding of cytokine crosstalk pathways in the tumor microenvironment, such as JAK/STAT, PI3K, AKT, Rac, MAPK, NF-κB, JunB, cFos, and mTOR, which are involved in angiogenesis, cancer proliferation and metastasis. Accordingly, targeting and blocking tumor-promoting cytokines or activating and amplifying tumor-inhibiting cytokines are considered cancer-directed therapies. Here, we focus on the role of the inflammatory cytokine system in pro- and anti-tumor immune responses, discuss cytokine pathways involved in immune responses to cancer and some anti-cancer therapeutic applications.

## 1. Introduction

Immunology is closely related to the complex discipline of oncology, where the immune response is recognized as a double-edged sword in terms of tumor progression, either by attenuating or promoting cancer invasiveness and metastasis. There are two distinct aspects of the immune response: (1) the innate, mediated by non-specific immunity cells, such as neutrophils, monocytes, macrophages, dendritic cells (DCs), natural killer (NK) cells, and γδT lymphocytes [1]; and (2) the adaptive, mediated by specific immunity cells, such as T (CD4^+^ helpers and CD8^+^ cytotoxic) and B lymphocytes, both working together to protect against pathogens [2] and abnormal cells. Immunocompetent cells (cells capable of mediating an effective immune response) include non-specific and specific immunity cells, but also non-immune cells, such as adipocytes. The latter can act as antigen-presenting cells (APCs) since they can express major histocompatibility complex (MHC) class I and II molecules and present their antigenic determinants to T cells to trigger an immune response, cytokine production, and subsequent intercellular signaling. In obese individuals, adipocytes can recruit and activate immune cells, such as adipose-resident T cells (ARTs) to stimulate the adipose pro-inflammatory response during the progression of obesity [3]. Naïve CD4^+^ T helper cells (Th) are activated after interaction with the antigen-MHC complex and differentiated into subsets of helper T cells, such as Th1, Th2, Th17, and regulatory T cell (Tregs) that produce different combinations of cytokines and other factors. Tregs are identified as those suppressing anti-tumor activity by inhibiting, among others, the differentiation of naïve CD4^+^ T cells into Th1, thus enhancing the tumor development and metastasis [4]. In addition, some immune cells play a dual role, such as neutrophils (cancer promoting—tumor-associated neutrophils (TANs)/suppressing) [5] and macrophages, divided into M1-like macrophages (pro-inflammatory and anti-cancer activities), M2-like macrophages (anti-inflammatory and pro-cancer activities) and tumor-associated macrophages (TAMs) [6].

These immune cells send signals to communicate with the microenvironment using cytokines that act as chemical messengers in an endocrine, paracrine, or autocrine manner and mediate intercellular communication of the immune system. Cytokines, called immunomodulatory agents, are synthesized under physiological and pathological conditions, and secreted by different cell types, such as immune cells, immunocompetent cells (e.g., adipocytes), and some cancer cells. They are involved in cellular (type 1) and antibody-mediated (type 2) immunity, as anti/pro-inflammatory and pro/anti-tumorigenic effectors depending on the microenvironment. Cytokines can affect the actions of other cells by binding to their surface receptors and subsequently activating numerous signaling pathways. There are different types of cytokines, including *chemokines*, *interleukins*, *adipokines*, *transforming growth factors* (*TGFs*), *tumor necrosis factor* (*TNF*), *colony-stimulating factors* (*CSFs*) and *interferons* (*IFN*), which can act alone, in synergic, protagonist, or antagonist manner to regulate inflammatory and immune responses [7].

*Chemokines* are chemoattractant cytokines that give chemical orders to attract inflammatory cells, such as leukocytes (monocytes, neutrophils) as well as other cell types, including endothelial and epithelial cells, to the site of interest [8]. They are classified as CX3C, CXC, CC, or C chemokines based on the positioning of conserved cysteine residues [9] and interact with G protein-linked transmembrane receptors called chemokine receptors [10]. Depending on their function, chemokines can be inflammatory (e.g., CXCL8, CCL3), recruiting cells through inflammatory stimulus, or/and homeostatic (movement and localization of cell subsets) [11].

Moreover, *interleukins* (ILs), low molecular weight cytokines, have both pro- and anti-inflammatory properties. They are secreted by almost all immunocompetent cells, such as (but not limited to) T cells, granulocytes, monocytes, macrophages, adipocytes, and endothelial cells [12]. They play essential roles in the development, differentiation, activation, maturation, migration, and adhesion of immune cells [13].

*Adipokines*, or adipocytokines, are cytokines specifically secreted by adipose tissue that is composed mainly of adipocytes, pre-adipocytes, macrophages, stromal cells, fibroblasts, and endothelial cells [14]. They include adipose tissue-specific cytokines (adiponectin, leptin), but also other cytokine types, such as interleukins, TNFs, or chemokines. Adipokines regulate energy expenditure, inflammation, appetite control, and fat distribution [15], but can contribute to an obesity-related low-grade inflammation, development of metabolic diseases [16], and also to cancer progression and metastasis [17]. There are two groups of adipokines based on their effect on the immune system: pro-inflammatory, such as leptin, TNFα, interleukin-1β (IL-1β), interleukin-6 (IL-6), and interleukin-8 (IL-8), potentially linking adiposity and inflammation, and anti-inflammatory, such as interleukin-10 (IL-10) and adiponectin [17,18]. Some of the adipokines participate in the anti-cancer activity, such as adiponectin [18] and some present tumorigenic properties, such as leptin [19].

*TGFs* are a subgroup of a larger family of protein hormones that are up-regulated in some human cancers and play several roles in growth and development of non-malignant and malignant cells. TGFα belongs to the epidermal growth factor (EGF) family, can induce epithelial development, cell proliferation, and is involved in tumorigenesis and angiogenesis [20]. It is secreted by M2 macrophages and various tumor cells, and can regulate T cells, NK cells, and macrophages in tumor microenvironment (TME), which contributes to the suppression of anti-tumor immunity and promotes tumor growth [21].

*CSF* has been implicated, for example, in breast carcinogenesis (varies by menopausal status) [22]. TNFα is a key cytokine involved in the generation of the proinflammatory response and many different cellular responses, such as the up-regulation of anti-apoptotic genes, inducing cell survival and proliferation, but also cancer invasion [23]. As an adipokine, it has an impact on the endocrine functions of adipose tissue and is associated with obesity, promotes insulin resistance and type 2 diabetes.

Finally, *IFN*, named after its ability to interfere with viral growth [24], is released by host cells and plays a controversial role in immune status modulation, anti-microbial/anti-viral host defense, up-regulation of antigen presentation and expression of MHC antigens. Type I IFNs (IFN-α, IFN-β) are produced by fibroblasts and monocytes when the body recognizes a virus that has invaded it. They stimulate the expression of proteins that will prevent the virus from producing and replicating its RNA and/or DNA. Type II IFN (IFN-γ) is released by CD8^+^ T and Th1 cells, activating cells, such as NK cells, M1-type macrophages and cytotoxic CD8^+^ T cells, increasing MHC I and II presentation, thus enhancing anti-tumor immunity [25].

So, cytokines affect the growth and function of many cell types and can activate or modulate specific or non-specific anti-tumor responses. In breast cancer, cancer cells are directly in contact with the adipose microenvironment and reciprocal interactions with immunocompetent cells, immune cells, infiltrating immune cells (TANs, M2, TAMs) [6], fibroblasts, cancer-associated fibroblasts (CAFs) [26], and endothelial cells, have been revealed, particularly in obese people [27]. The TME could play a critical role in all stages of tumor development, contributing to the development, progression, and metastasis of malignant cells. Some cytokines (leptin, IL-1β, IL-6, IL-8, IL-23, IL-17, TGF-β, IL-10) are mostly reported to stimulate while others (Il-2, IL-12, IFNs) inhibit breast cancer proliferation and/or invasion. The former contribute to the establishment of a tumor-promoting inflammation, recognized as a hallmark of cancer [28]. There are two types of inflammation, acute inflammation induced by common bacterial infections and viruses, and chronic inflammation associated with chronic disease, obesity [29], and cancer [30]. To some extent, inflammatory environment and chronic inflammation have long been associated with increased incidence of malignancy and tumor-promoting effect [31,32] through bioactive molecules. Therefore, using various agents to activate or boost the immune system and attack cancer cells by natural mechanisms through immunotherapy is becoming a powerful clinical strategy to treat cancer [33]. To this end, it is extremely relevant to understand the regulatory mechanisms of inflammation in breast carcinogenesis and metastatic progression to enable identification of novel therapeutic targets for tumors [28].

In this review, we are discussing the relationship between cytokines recognized as having an important role in TME in breast cancer, such as adipokines and some other cytokines, and how they regulate cancer immunity. It provides a comprehensive overview of cytokine crosstalk, their biological roles, the signaling pathways and transcription factors responsible for the anti- or pro-tumor response and suggested immunotherapeutic strategy for each cytokine for effective treatment of cancer. The details of cytokine overview are described in Table 1.

## 2. Leptin

Leptin is an obesity-associated adipokine known as the “obesity hormone”, whose circulating levels increase proportionally to body fat mass [193]. It is a pleiotropic cytokine of the IL-6 family produced and expressed mainly by adipose tissue [194], enterocytes [195], but also by immune cells, endothelial cells [196], fibroblasts, CAFs [197], and some tumor tissues. It is encoded by the obese (Ob) gene which maintains energy homeostasis, through a central feedback mechanism at the hypothalamus. Under normal physiological conditions, leptin plays a key role in the long-term regulation of energy balance by reducing appetite and increasing the metabolism, thereby controlling food intake and body weight [36,37]. In obese individuals with insulin resistance, high levels of leptin have been correlated with the amount of fat in the body, acting as a pro-inflammatory cytokine and amplifying the insulin resistance process [40,198,199].

Leptin promotes cell proliferation and the development of breast cancers [200] and may be a potential biomarker of breast cancer risk in women, especially overweight/obese or postmenopausal women [201]. Women with elevated serum leptin have a higher risk of breast cancer [202]. Higher levels of leptin have been observed in the invasive stage, from mammary ductal carcinoma in situ to invasive ductal carcinoma [27], in the cytosol and nuclei of metastatic cells [203,204].

In addition, leptin promotes epithelial-to-mesenchymal transition (EMT) [39] by up-regulating the expression of CSC/EMT-related genes [205], tumor evasion, cell migration [206,207], tumor size, overexpression of estrogen and progesterone receptors [208], protection of malignant cells from apoptosis [38], and breast cancer metastasis [209]. The functions of leptin are enhanced by the cross-talk with multiple cytokines. Indeed, it is an important mediator of interactions between breast cancer cells and TAMs [38] by stimulating the production of IL-8 which is directly involved in tumor growth [210,211].

The molecular actions of leptin are mediated by ObR (leptin receptor), a large membrane protein member of the class I cytokine receptors [212]. Following leptin binding to its receptor (Lep/ObR), Janus kinase 2 (JAK2) becomes activated through autophosphorylation, which in turn phosphorylates tyrosine residues on the intracellular domain of the receptor for the signal transducer and activator of transcription 3 (STAT3). STAT3 forms dimers and translocates into the nucleus where it activates the transcription of different target genes involved in cellular activation, proliferation, and differentiation [213]. Phosphorylated JAK2 also activates other pathways, including mitogen-activated-protein kinase (MAPK)/extracellular signal-regulated kinase (ERK) promoting cell proliferation and differentiation [214,215] and phosphatidylinositol 3-kinase (PI3K)/protein kinase B (Akt)/Rac that mediates the regulation of cell cycle, growth, proliferation, and energy metabolism [34,35,216]. It also activates the hypoxia-inducible factor-1α (HIF-1α) and NF-κB to regulate the vascular endothelial growth factor (VEGF) and then promotes angiogenesis in mammary tumors [35]. In addition, it activates reactive oxygen species (ROS) production in human epithelial mammary cells [217], and can regulate metabolic reprogramming to promote cellular growth [218] (Figure 1 and Figure 2).

Leptin-targeting drugs can affect different cell types, including endothelial cells, adipocytes [219], cancer cells with a less aggressive phenotype, and immune cells by decreasing macrophage recruitment, phagocytic activity, and cytokine production [209]. Thus, the use of antibodies targeting leptin/Ob-R able to perform an antagonist activity could be used in breast cancer therapy. For example, researchers blocked leptin signaling by a leptin peptide receptor antagonist that simultaneously decreased VEGF/VEGFR2 and IL-1 levels [220], or by producing a synthetic farnesoid X receptor (FXR) (regulator of the dialogue between breast cancer cells and cancer-associated fibroblasts) agonist GW4064, which affects the tumor-promoting activities of CAFs in breast malignancy [221]. Some new super active leptin antagonists (D23L/L39A/D40A/F41A) exhibited more than 60-fold binding to the leptin receptor [222]. In breast cancer, a leptin antagonist-honokiol, a bioactive polyphenol from *Magnolia grandiflora*, is reported to activate the liver kinase B1-miR-34a axis and finally inhibit EMT [223].

## 3. TNFα

TNFα is a major inflammatory cytokine and adipokine described as a circulating and endotoxin-induced factor of hemorrhagic tumor necrosis when present in high concentrations. It is mainly produced by macrophages and in a lower extent by B and T (cytotoxic CD8^+^ and CD4^+^ Th1) cells, NK cells, mast cells, fibroblasts, osteoclasts, endothelial, and muscle cells. It can act both in autocrine and paracrine signaling [224]. In general, as an adipokine, it is associated with obesity, promotes insulin resistance and type 2 diabetes. By regulating many cellular and biological processes, it may also have an impact on the endocrine functions of adipose tissue. It can reduce adiponectin secretion, inhibit carbohydrate metabolism, lipogenesis and adipogenesis, and stimulate lipolysis [225]. As a cytokine, it is released by macrophages to alert other cells of the immune system as part of an inflammatory response [226]. Thus, it is involved in the generation of a pro-inflammatory response, as it can activate and up-regulate more than 400 inflammatory genes and stimulate many different cellular responses, such as transcription of pro-inflammatory genes of inflammatory cytokines and chemokines [48,49].

TNFα has a key role in breast cancer. It is involved in breast cancer cell survival and/or proliferation [52], tumor-promoting, aggressiveness [53], tumor-promoting macrophage infiltration, CAF phenotype, inflammatory chemokine expression, such as C-X-C motif ligand 8 (CXCL8 = IL-8), and angiogenesis [50]. The prometastatic role of TNFα and its involvement in the EMT process required tumor cell migration to establish breast cancer metastasis [41] and its over-expression has often been associated with aggressive cancer behavior and poor prognosis [227]. TNFα and IL-1β are the essential pro-inflammatory cytokines often found in TME. In breast cancer, inflammation is known to be associated with poor prognosis and higher risk of recurrence in patients [228]. TNF-α production by peripheral blood T cells in patients with inflammatory breast cancer was positively correlated with the detection of circulating tumor cells expressing EMT markers [229]. TNFα is recognized by a variety of stromal cells, mainly by TAMs, adipocytes, epithelial and malignant cancer cells themselves [41,42,43,44]. In addition, TNFα and IL-6 have been positively correlated with aromatase activity in human breast adipose tissue in primary culture by increasing aromatase mRNA [230].

Ultimately, TNFα binds to two different receptors, TNFR1 and TNFR2, and is capable of activating NF-κB, c-Jun, activating protein-1 (AP1), MAPKs, Akt, TAZ, and c-Jun N-terminal kinase (JNK)/P38 [45,46,47]. TAZ is a transcriptional coactivator, activated via NF-κB to induce tumor initiation and self-renewal in breast cancer stem-like cells, a subpopulation of primary breast tumor cells with differentiation and self-renewal capacities implicated in tumor generation, cancer relapse, and metastasis [231]. TNFα expression is transcriptionally induced by NF-κB, c-Jun, AP1, and nuclear factor associated with activated T cells (NFAT). Ligand-occupied TNFR1 induces NF-κB and c-Jun activations, which inhibit apoptosis, increase transcription of survival factors (Bcl-2), transcription of pro-inflammatory genes, cell death pathways through apoptosis (apoptosis signaling kinase 1 ASK1) or necroptosis [232], promote transcription of EMT-related factors (i.e., matrix metallopeptidase-2 [MMP2], MMP9), and decrease E-cadherin transcription. However, TNFR2 can activate NF-*κ*B through a pathway similar to that of TNFR1, but it can also activate endothelial/epithelial tyrosine kinase phosphorylation that can activate the Akt pathway. In particular, prolonged exposure of breast cancer cell lines to TNFα induces EMT through the activation of IKKβ and NF-κB [233], cancer cell migration via the MAPK/ERK signaling pathway [234], and MMP2 and MMP9 expression [235,236]. TNFα has been shown to induce the gene expression of aromatase in undifferentiated adipose fibroblasts via c-Fos and c-Jun, and inhibits adipogenic differentiation in breast TME [237]. However, IL-10, through inhibition of TNFα induction, can suppress aromatase mRNA expression in human adipose tissue [238] (Figure 1 and Figure 2).

TNFα can induce resistance to breast cancer therapies, such as ionizing radiation therapy [239] and chemotherapy [240]. In breast cancer, TNFα inhibition has been shown to increase the sensitivity to doxorubicin [241]. TNFα mediates TNFR1-dependent IL-17 production by CD4^+^ T cells [53] and can promote the activities of immune and cancer cells within tumors. Targeting TNFR2 with antagonistic antibodies (anti-PD-1) has been shown to inhibit the proliferation of cancer cells and tumor-associated Tregs [242]. Therefore, blocking TNFα or its receptors may have significant anti-tumor effects [31,243,244] and inhibition of TNFα on tumorigenesis can be positive depending on the immune microenvironment and TNFα concentration.

## 4. Interleukin-1**β** (IL-1**β**)

IL-1β, known as an endogenous mediator of leukocytes, is a potent pro-inflammatory cytokine involved in the modulation of autoimmune inflammation [245], proinflammatory response, cell proliferation, and cell differentiation. IL1-β gene transcription in macrophages is mainly induced by lipopolysaccharides (LPS) via toll-like receptors (TLRs) or IL-1β itself and by TNFα via the TNF receptors, which are capable of activating NF-κB and subsequently STAT1 pathway [246]. This cytokine is primarily produced by activated macrophages as a proprotein called pro-IL-1β, an inactive precursor. It is proteolytically processed to its active form IL-1β by inflammatory caspase 1 cleavage (CASP1/ICE) [247] and subsequently binds to IL-1 receptor CD121a/IL1R1 and CD121b/IL1R2 subunits [248,249]. It can also be produced and secreted by a variety of cell types, such as adipocytes, monocytes, DCs, fibroblasts, B cells, TAMs [54,55], and some cancer cells to enhance tumor-promotion [54,56].

In the TME, IL-1β is up-regulated in many solid tumors, including breast cancers [54]. IL-1β has been reported to control tumor invasion, up-regulate the initiation and development of primary tumor, increase the aggressivity of luminal breast cancer cells, and increase IL-6 production through NF-κB pathway [250] leading to tumor growth and aggressiveness. IL-1β has been linked with poor prognosis in breast cancer [251] and plays a critical role in the recruitment and maturation of adaptive T cell-mediated immunity including CD4^+^ and CD8^+^ T cells [252] and myeloid cells [68]. IL-1β also activates focal adhesion kinase and Src to induce MMP9 production and invasion of MCF-7 breast cancer cells [253]. In vitro studies have recently shown that primary breast cancer cells cocultured with monocytes exhibit increased IL-1β, IL-8, and MMPs [254], suggesting that inflammation and subsequent recruitment of immune cells promote breast cancer development at early stages. Furthermore, in a spontaneous in vivo model of breast cancer, cancer cells stimulate systemic inflammation by producing IL-1β, which leads to stimulation of IL-17 production by γδT cells [57] and can inhibit anti-tumor CD8^+^ T cell activity [57]. TAM-derived IL-1β can increase cyclo-oxygenase-2 (COX-2) expression in breast cancer cells and contribute to cancer progression [60,61], migration/invasion [65], regulate metastatic process [255], and EMT [66].

M2-type macrophages, TAMs and cancer cells can stimulate IL-1β production [256] and subsequently, IL-1β increases macrophage recruitment through the expression of monocyte chemoattractant protein (MCP)-1, which can be polarized into TAM and promote cancer, tumor growth, and metastasis [257]. M1-type macrophages are the cells most frequently induced to produce IL-1β, and subsequently stimulate proinflammatory mediators [258], the production of angiopoietin-like 4 (ANGPTL4), and VEGF-A in adipocytes [259], MMP9 [59], and matrix-remodeling activities [69,70]. In this context, obesity induces secretion of MCP1 and IL-1β by adipocytes associated with breast tissue, which increases macrophage recruitment and the formation of crown-like structures (CLS), followed by the secretion of CXCL12, a key effector responsible for stromal vascularization and angiogenesis [260]. In addition, it promotes activation of PI3K/Rac 1-regulated, ERK1/2 and AP-1, and the reorganization of the actin cytoskeleton in invasive cancer cells [58] (Figure 1 and Figure 2).

Overall, IL-1β represents a major upstream cytokine. Inhibition of IL-1β signaling in malignant tumors are now considered as potential targets for cancer therapy. Anti-IL-1β therapy is shown to decrease metastasis [66], invasiveness, inflammation-mediated immunosuppression concomitantly with increased anti-tumor immunity response [261]. In breast cancer, a variety of drugs are currently used in clinical practice to target the IL-1 signaling pathway, including anakinra (Nbib1802970), the standard of care used for phase I metastatic breast cancer, and PDR001 (Nbib2900664), in combination with CJM112, EGF816, Ilaris (canakinumab), or Mekinist (trametinib) used for triple-negative phase I and phase II breast cancer [262].

## 5. Interleukin-6 (IL-6)

IL-6 is a multifunctional cytokine and an adipokine that clearly has both tumor-promoting and pro-inflammatory effects [75,76]. It was originally identified as B cell stimulating factor-2 (BSF-2) and the inducer of immunoglobulin production [263].

IL-6 is produced by non-malignant cells, such as monocytes, macrophages, T cells, B cells, fibroblasts, endothelial cells, and adipocytes [75,76,77]. IL-6 transcription is induced by various stimuli, such as TNFα and ROS [264,265]. It promotes Th2 differentiation and simultaneously inhibits Th1 polarization, maintains dynamic balance between Th1 and Th2 immune cells [266], and regulates the balance between IL-17-producing Th17 cells and Treg [267]. In the TME, its main sources are some cancer cells [78,79], TAMs [71], myeloid-derived suppressor cells (MDSC), Th2 cells, and CAFs [73,74].

IL-6 is a major player in chronic inflammatory diseases, autoimmune diseases, cancer, and tumor immunity [268,269]. In breast cancer, IL-6 is found to be overexpressed [270] and has been described as a tumor-promoting cytokine through its major effector STAT3. It is a key factor in malignancy by promoting cell growth (via the suppression of apoptosis and promotion of angiogenesis [271]), macrophage polarization [72], tumor initiation, progression, and metastasis [31,268,272], by selectively recruiting mesenchymal stem cells to sites of carcinoma growth where they interact with breast cancer stem cells [273]. IL-6 controls cancer stem cell renewal and induces cancer cell migration [274], and EMT [275]. IL-6 stimulates aromatase and therefore estrogen biosynthesis, thus contributing to hormone-dependent breast cancer [276]. CAFs in breast tumors express high levels of IL-6 and subsequently mediate epithelial-stromal interactions to promote tumorigenesis [277]. High IL-6 levels in breast cancer tissues maintain the aggressive phenotype [278].

The IL-6 receptor (IL-6R) is a hetero-trimeric complex receptor composed of an IL-6-binding receptor molecule α-subunit (gp80) and two signal transducing β-subunits of glycoprotein 130 (gp130) [264]. Classical IL-6 signaling is initiated by its binding to IL-6Rα (present either on the cell surface or in secreted form, which then induces *cis* dimerization of gp130, auto- and trans-phosphorylation and activation of the associated JAK1, JAK2, and tyrosine kinase 2 (TYK2) [81]. The tyrosine residues phosphorylated by JAKs in the intracellular domain of gp130 activate the transcription factors STAT3/STAT1 [82,83], Src homology 2 (SH2)-containing protein tyrosine phosphatase 2 (SHP2), ERK, MAPK, and PI3K signaling pathways [84,85]. Thus, STAT3 may mediate tumorigenesis by increasing cell-cycle progression [87], resistance to apoptosis (inducing the expression of BCL2, BCL-XL, and survivin) [279,280], metastasis [89], and senescence [86]. In addition, IL-6/STAT3 signaling activates the transcription of target genes, including the proto-oncogenes c-myc, JunB, cFos, and mTORC1 metabolic regulators [279,281,282,283,284,285]. It also blocks DC differentiation, thereby preventing T cell activation and inducing T cell death [91]. Activation of STAT3 in tumor-associated endothelial cells, TAMs, and cancer cells, induces their ability to express basic fibroblast growth factor (bFGF) and VEGF, promoting rapid vascularization. Indeed, IL-6 may up-regulate circulating VEGF in breast cancer patients and promote angiogenesis and metastasis [286]. IL-6 may promote the mobilization of anti-tumor CD8^+^ effector T cell responses, the development of APC, such as DCs and cytotoxic T cells [91,93], as well as the survival, proliferation, differentiation, and recruitment of leukocytes [94,95]. Activation of IL-6Rα expressed on DCs can act directly on CD4^+^ T cells through gp130 and subsequently induce Th17 cells [92]. Furthermore, it has been shown that high expression of IL-6Rα induces resistance to apoptosis in breast cancer [287] (Figure 1 and Figure 2).

Therefore, blocking IL-6 (i.e., anti-IL-6 therapy), targeting its receptor in combination with signaling from other anticancer therapies, and targeting the IL-6-STAT3 axis is a potential therapeutic strategy that may be beneficial in the treatment of breast cancer [206,288,289]. Down-regulation of IL-6 is linked to a better response to breast cancer treatment [290] and a recent study showed that the IL-6R neutralizing antibody, tocilizumab, abrogated IL-6 signaling in breast cancer [291]. A synthetic anti-gp130 compound (bazedoxifene) inhibited IL-6-induced growth of breast cancer cell lines and down-regulated STAT3 phosphorylation [292]. An IL-6R antagonist (Tocilizumab) for trastuzumab-resistant HER2-positive metastatic breast cancer is being evaluated in a clinical trial (NCT03135171) [293].

## 6. Interleukin-8 (IL-8)

IL-8, also known as CXCL8 [294], is a proinflammatory chemokine, produced and expressed by monocytes [98], macrophages [96], neutrophils [103,104], T cells, fibroblasts [99], epithelial cells [100], and vascular endothelial cells [101]. It is known as neutrophil chemotactic factor inducing neutrophil and granulocytes recruitment, degranulation [295], and phagocytosis [296]. IL-8 has been associated with proinflammatory response in obesity [297] and cancer [298] and can induce the recruitment of inflammatory cells and oxidative stress mediators into localized inflammation to exert cytotoxic activities [120].

IL-8 secretion by TAMs [97], CAFs [102], and some cancer cells [105] promotes their recruitment, proliferation, and survival, suppresses CD8^+^ T cell activity [119], and stimulates cellular secretion of additional growth factors that contribute to breast cancer progression. Several studies have shown that in breast cancer cell lines, invasion is directly proportional to IL-8 expression [299] where, various cytokines, such as IL-1β, TNFα, and IL-6, as well as hormones, such as progesterone and estrogen, are thought to up-regulate IL-8 expression in breast cancer cells compared to normal breast tissue [300]. Breast cancer cells are reported to secrete IL-8, express CXCR1/2, and promote breast cancer initiation and progression, as well as tumor cell migration and invasion [301].

Cellular responses to IL-8 are mediated by two cell-surface G protein-coupled serpentine receptors for a group of C-X-C chemokines, termed CXCR1 and CXCR2 [302]. CXCR1 is specific for IL-8, whereas CXCR2 can mediate cellular responses [303] after binding other chemokines as well, such as IL-2 and IL-6. IL-8 expression can be regulated by a variety of stimuli, including inflammatory signals (e.g., TNFα, IL-1β) [99,100], environmental and chemical stresses (e.g., hypoxia or exposure to chemotherapy agents), and steroid hormones [304]

IL-8 signaling can activate and regulate PI3K (in neutrophils) [106], PKB/Akt [106], MAPK (cell proliferation in neutrophils [103,104], endothelial [305], and in some breast cancer cell lines) and the Raf-1/MAP/ERK1 cascade, p38 MAPK signaling cascade [107] and phospholipase C [306,307]. In addition, it can activate Src, focal adhesion kinase FAK [109], STAT3 [97,110], JAK2/STAT3/Snail (in cancer cells) [96], signaling kinases correlated with cancer progression. Thus, it forms an immunosuppressive microenvironment to enhance tumorigenicity, cancer cell motility (via MMP9 expression [308], PI3K-Akt signaling, and E-cadherin down-regulation) [309], EMT [118,310], invasion and metastasis [115]. In addition, it can activate endothelial cells in the tumor to promote vascular endothelial cell proliferation [311] and angiogenesis [312,313,314]. IL-8 receptors on endothelial cells can activate Ras/PI3K [111,112], NF-κB/VEGF activation [113], chemoresistance [101], and metastasis [315] (Figure 1 and Figure 2).

The multiple effects of IL-8 signaling on different cell types present in the TME suggest that targeting CXC chemokine signaling (including, but not limited to IL-8) may have important implications for halting disease progression and helping to sensitize tumors to chemotherapeutic and biologic agents [108]. Furthermore, a clinical study has shown that elevated serum IL-8 levels correlate with higher-stage breast tumors [316], invasiveness and poor prognosis [299,317]. CXCL8-CXCR1/2, as a key driver of immune suppression, may interfere with the differentiation and function of stromal and immune cells in the TME, ultimately affecting immunotherapy [318,319]. Therefore, blocking the CXCL8-CXCR1/2 axis by using small molecules or antibodies were testes, such as SB225002 (CXCL8 inhibitor binding to CXCR2) [320] to inhibit tumor progression in HER2+ breast cancer [321], and danirixin (GSK1325756) to suppress cancer migration, invasion, and metastasis. Finally, various orally active small-molecule non-competitive antagonists of CXCR1 and CXCR2, such as SCH527123 (Merck) [322], repertaxin (Dompé, Milan, Italy), and SCH479833 (Merck, Whitehouse Station, NJ, USA), have demonstrated anti-tumor effects in breast cancer xenograft models [323].

## 7. Interleukin-17A (IL-17A)

IL-17A is the founding member of the IL-17 family of pro-inflammatory cytokines associated with allergic responses [324]. In humans, increased IL-17A level is associated with infections, antimicrobial immunity, chronic inflammatory diseases, obesity, and autoimmune diseases [325,326]. Following stimulation of CD4^+^ T cells by cytokines, such as IL-6, TNFα, IL-21, IL-23, TGF-β, and IL-1β, a naïve CD4^+^ T cell differentiates into an inflammatory class of Th17 cells [130,131] and secretes IL-17A. It is primarily produced by a group of Th cells known as Th17 cells [132], in response to their stimulation by IL-23 [327]. It is also produced by other cells, such as CD8^+^ T cells, γδ T cells, NKT, and NK cells [328]. Both STAT3 and NF-κB signaling pathways are required for this cytokine-mediated IL-17 production [130,329]. Additionally, the RAR-related orphan nuclear receptor-γt (RORγt) is the most specific transcription factor promoting Th17 cell differentiation [330]. The most notable role of IL-17 is its involvement in the induction and mediation of proinflammatory responses under inflammatory conditions through the stimulation of many cell types (macrophages, fibroblasts, endothelial cells) to produce other cytokines, such as TNFα, IL-1β, IL-6, CSF, TGF-β, and chemokines, including IL-8 to amplify the inflammatory response [326,331,332].

Biologically active IL-17A interacts with the type I single-pass transmembrane protein, IL-17R cell surface receptor. It can be expressed by T cells [333], macrophages, fibroblasts, endothelial, epithelial, and cancer cells. Following binding to its receptor, IL-17 activates several signaling cascades that, in turn, lead to the induction of chemokines and the recruitment of immune cells, such as neutrophils and monocytes to the site of inflammation. It stabilizes RORγt important pathways that have become a focus around the IL-17 cytokine family, including NF-κB [334], and the production of IL-6 and TGFβ, and IL-1β signaling pathways that induce the expression of several proinflammatory cytokines.

High levels of IL-17A in the breast microenvironment are associated with the highly invasive and aggressive phenotype of breast cancer. IL-17A induces activation of ERK1/2 phosphorylation, p38 MAPK and STAT3 (IL-6-STAT3) signaling pathways, and promotes tumor growth [133], tissue invasion, tissue-remodeling and matrix degrading substances (such as MMPs, including MMP2 and MMP9) [335], migration [134], inhibition of apoptosis, and angiogenesis [336] (via activation of VEGF and CXCL8 expression [57]). IL-17A can stimulate breast tumorigenesis and, in turn, CAFs, γδT and breast tumor cells increase Th17 cell recruitment and IL-17A production [337]. Tumor-derived IL1β activates γδ T to produce high levels of IL-17, which leads to neutrophil expansion and altered neutrophil phenotype. Neutrophils produce inducible nitric oxide synthase (iNOS), which inhibits the activity of anti-tumor CD8^+^ T-cells and subsequently stimulates cancer progression [338], migration, invasion, and metastasis [339]. Interestingly, IL-17 increases expression of MMP-9, indoleamine 2,3-dioxygenase, COX-2, and MMP-13. furthermore, IL-17 up-regulates the gene expression corresponding to M2-type TAMs. IL-17 acts alone or in synergy with other stimuli (such as TNFα and IFN-γ) to activate the expression of many genes, including cytokines, such as IL-6, IL-19, IL-20, IL-24, TNFα, and granulocyte-CSF, chemokines, such as IL-8, CXCL1, CXCL2, CXCL5, CXCL9, CXCL10, C-C motif ligand 2 (CCL2), CCL7, and CCL20 [326,332,340,341], MMP13, receptor activator of nuclear factor kappa-B ligand (RANKL), and antimicrobial peptides (lipocalin 2, β-defensin-2, S100A7, and S100A8/9) [326] (Figure 1 and Figure 2).

A better understanding of the character of Th17 cells in tumor immunity should generate opportunities for the progression of new therapeutic approaches for cancer patients. Therefore, different strategies should be used depending on the type of cancer and the clinical influence of IL-17 in tumor development. For example, the use of neutralizing antibodies against Th17-related cytokines has been shown to decrease EMT and significantly inhibit progression and metastasis of lung cancer [342] and control invasive breast tumors [343]. Similarly, because the absence of γδ T cells or neutrophils profoundly reduces metastasis without influencing primary tumor progression, regulation of the Treg/Th17 axis, inhibition of the γδ/IL-17/neutrophil axis, and blockade of IL-17RB in cancer cells could be an effective therapeutic approach in breast cancer [57,344].

## 8. Interleukin-23 (IL-23)

IL-23 is a heterodimeric proinflammatory cytokine that belongs to the IL-12 cytokine family. These two share a common p40 subunit that is covalently linked either to p35 subunit to form IL-12 or to p19 subunit to form IL-23 [345]. It is a key cytokine for the differentiation, maintenance, and expansion of Th17 cell, as discussed earlier [345,346]. IL-23 is mainly produced by DCs and macrophages, but also by monocytes, neutrophils, innate lymphoid cells [121,122,123], γδ T cells, B cells, and M2/TAM cells [347]. The heterodimeric IL-23 cytokine receptor is composed of IL12Rβ1 (receptor for IL-12) that signals through tyrosine kinase-2 (TYK2), and IL-23R that signals through JAK2 to activate STAT3 [124,125]. IL-23 receptors can be expressed in T cells, NK cells, NKT cells, tumor cells and are weakly expressed on monocytes, macrophages, and DC populations [124].

Within the immune and inflammatory microenvironment, IL-23 participates in the progression of chronic inflammation and its maintenance and plays a critical role in autoimmune diseases (via IL-23/IL-17 immune axis) [123,348,349]. Notably, it can manipulate host immune responses and modulate TME cells. IL-23 is a key cytokine for the differentiation, maintenance, and expansion of Th17 cells via activation of STAT3, which in turn stabilizes RORγt, as previously detailed.

It has been identified as a link between tumor-associated inflammation and tumor immune evasion [129], and can directly affect a variety of premalignant and malignant tumors. IL-23 is involved in breast carcinoma cell metastasis [349] and is associated with higher breast cancer tumor size and stages [350]. The pro-tumorigenic role of IL-23 was first reported by Langowski et al., where its genetic blockade resulted in increased cytotoxic T cell tumor infiltration [129]. It has been shown to activate the NF-κB signaling and then induces immune cell activation which exacerbates inflammation and promotes tumor growth. IL-23 decreases infiltration of Treg and CD8^+^ T cells [126] to promote the infiltration of M2-type macrophages and neutrophil cells and their overexpression and secretion of pro-tumor immunosuppressive cytokines [126], such as TGF-β and IL-10. Furthermore, IL-23 increases the expression of endothelial and angiogenic markers, such as VEGF [129,351], MMP9, CD31, and the proliferative marker Ki67 in tumors [126,352]. In turn, mammary tumor cells produce IL-6, VEGF, CCL22 to recruit TAMs, which stimulate IL-23 production and maintain the suppressive activity of Treg in the TME [353] (Figure 1 and Figure 2).

In conclusion, IL-23 has been shown to play multifunctional roles in tumorigenesis by inhibiting anti-tumor effector immunity. It provides an important molecular link between the tumor-promoting inflammatory response and the failure of adaptive immune surveillance to infiltrate tumors [354]. Therefore, injectable IL-23 inhibitor drugs, such as guselkumab/Tremfya, risankizumab-rzaa/Skyrizi and tildrakizumab-asmn/Ilumya [355], blocking-up the process of IL-23 (M2 and neutrophils), neutralizing antibodies specific to IL-23p19 (G23-8 antibody can down-modulate MMP9 expression and increased surveillance of CD8^+^ T cells), and IL-23p19 antagonists can provide effective anti-tumor therapy [126,129,356,357].

## 9. Interleukin-12 (IL-12)

IL-12 is a member of a small family of heterodimeric pro-inflammatory cytokines [345], produced and expressed primarily by APCs, such as DCs and activated macrophages, depending on the immune context [356,358]. Despite its similarities to IL-23, IL-12 may be able to stimulate effector cells of innate and adaptive immunity, activate macrophage polarization to M1 type, and enhance anti-tumor cytotoxic immune responses in the TME [135]. The IL-12 receptor comprises IL-12Rβ1 and IL-12Rβ2 subunits and is commonly expressed in T cells, NK cells, NKT cells, monocytes, macrophages, and DC populations [124].

Following binding of IL-12 to its receptor expressed on target cells, phosphorylation and homo-dimerization of STAT4 are promoted by JAK2 and TYK2 [135]. Therefore, activated STAT4 in CD4^+^ T cells induces transcription of the transcription factor T-box (T-bet) that can control IFN-γ production by Th1 cells [359], whereas in combination with STAT4, it enhances transcription of IL-12Rβ1, regulates Th1 cell differentiation and promotes expression and activation of Th1-associated receptors [360,361].

IL-12-mediated Th1 activation releases Th1-specific cytokines, including IFN-γ, to solicit and recruit cytotoxic NK and CD8^+^ T cells. IL-12 promotes the differentiation of naïve CD8^+^ T cells to the effector phenotype and acts as a CD8^+^ T-anti-apoptotic factor that can directly destroy microorganisms and cancer cells through the release of perforin and proteolytic enzymes [362,363]. Overall, IL-12 targets and modulates Th1, CD8^+^ T, NK, and APC cells that regulate immune surveillance, effective antimicrobial and cytotoxic activity [360,364]. Thus, it promotes efficient anti-tumor responses [360,365], cancer cell elimination, and tumor clearance in many cancers, such as breast cancer [366], through inhibition of the PI3K/AKT/mTOR signaling pathway. In addition, T-bet and STAT4 down-regulate RORγt, which limits the generation and proliferation of Th17 and Treg within the tumor [367,368]. Furthermore, it up-regulates MHCI on tumor cells to facilitate self-antigen presentation, increases the production of chemokines, such as CXCL9, CXCL10, CXCL11, and IFN-γ to attract effector immune cells, such as CD8^+^ T, NKs, and M1 macrophages [369,369]. In addition, IL-12 can increase the infiltration of IFN-γ-producing NK cells where, mechanistically, activation of IFN-γ signaling inhibits the proliferation of Tregs and subsequently converts them into IFN-γ-producing T cells [370]. Thus, it enhances the body’s immune response against cancer [371], promotes cancer cell apoptosis, increases MHCI expression on APCs, and down-regulates intratumor VEGF [372] (Figure 1 and Figure 3).

Therefore, IL-12 can be considered a strong candidate for immunotherapy-based interventions. It may be beneficial in controlling tumor growth by activating effective anti-tumor cytotoxic immune responses and killing tumor cells by tumor-specific cytotoxic NK and CD8^+^ T cells [373,374]. Recombinant human IL-12 (rhIL-12) is currently in clinical trials for the treatment of cancer and may also play a beneficial role in synergy with chemotherapy (through activation of NK cells) and radiation therapy (reducing complications) [356,375,376]. Various therapies have been devised. For advanced solid tumors, such as breast, a mRNA-based IL-12 delivery (SAR441000), an mRNA mixture encoding IL-12sc, interferon alpha2b, GM-CSF, and IL-15sushi/ is being evaluated [374].

## 10. Interleukin-2 (IL-2)

IL-2 is a pro-inflammatory cytokine, doted of the anti-tumor response [377]. It was firstly identified by Morgan and colleagues in 1976 as “T-cell growth factor” (TCGF) [378]. Among other things, IL-2 can modulate the differentiation of CD4^+^ T cells into Th1 and Th2 [147], increase the cytolytic activity of NKs, induce the development of Treg and cytotoxic T cells (CD8^+^ T cells), while it can both inhibit the differentiation of Th17, but also stimulate their expansion [148]. In addition, IL-2 promotes autocrine survival and cytolytic activity of T and NK cells in anti-tumor immunity [379]. The main sources of IL-2 are antigen-stimulated CD4^+^ T cells, but it can also be produced by activated CD8^+^ T cells [136], activated DCs, and NK cells [137,138].

IL-2 exerts its effects by binding to the IL-2 receptor subunits, IL-2Rα (CD25), IL-2Rβ (CD122), and IL-2Rγ (CD132), with different affinities [380,381]. The α chain is unique to IL-2 and binds it with low affinity, without transducing a signal because of its short intracellular chain [382]. IL-2Rβ is the key component of the IL-15 receptor, whereas the γ chain is shared by the receptors for IL-2, IL-4, IL-7, IL-9, IL-15, and IL-21 [136]. Notably, the α chain functions to initially bind IL-2, which localizes it to the cell surface, effectively increasing its concentration and also inducing a conformational change in IL-2R. Heterodimerization of the intermediate (IL-2Rβγ) and high affinity (IL-2Rαβγ) [383] receptor is essential for the activation of several IL-2 transducers and T cell signaling [384]. IL-2Rα can be expressed by naïve T cells (can be triggered rapidly by T cell receptor TCR), activated CD4^+^ and CD8^+^ T cells, mature DCs, B cells, and endothelial cells [144,145,146]. In addition, Tregs and some NK cells can also express high levels of the α chain after the IL-2 stimulation [385], allowing them to consume IL-2 more efficiently than CD4^+^ and CD8^+^ effector, even at low levels [386] (Figure 2). The IL-2Rβ is mainly expressed by Treg, memory CD8^+^ T cells, and NK cells. The intracellularly stored IL-2Rγ subunit is expressed primarily by hematopoietic cells [387], some tumor cells [143], and by CD4^+^ T only during activation [388], whereas the dimeric IL-2Rβγ receptor is expressed by memory CD8^+^ cells, naïve, T, and NK cells. Tregs and activated T cells express high levels of trimeric IL-2Rαβγ complex [387].

Further binding of IL-2 to its receptor, IL-2Rβγ or IL-2Rαβγ complex, results in activation of the JAK1 and JAK3, which activate the recruitment and phosphorylation of STAT transcription factors, primarily STAT5, but also STAT1 and STAT3. Subsequently, three major signaling pathways, including PI3K-AKT, JAK-STAT, and MAPK/ERK are activated [149] to mediate cell growth, survival, and differentiation, and cytokine production (IL-4, IL-6, IL-12) [150,151]. Thus, in breast cancer [389], IL-2 plays an anti-tumor effect [390] due to increased recruitment of IL-2-releasing NK cells and the induction of anti-apoptotic function in CD8^+^ T cells [391], the main effector of the antitumor response [392]. IL-2 can promote activation and proliferation of CD8^+^ T cells in early tumor stage but can deplete CD8^+^ T cells in late tumor stage [393] (Figure 1).

The multiple effects of IL-2 signaling on the different cell types present in the TME may be beneficial in developing T cells and controlling tumor growth by activating effective antitumor cytotoxic immune responses and killing tumor cells. Thus, IL-2 administration and adoptive transfer of antitumor T cells cultured with IL-2 have represented highly effective therapies in patients with solid human cancers, such as metastatic renal cancer, and melanoma [33,386].

## 11. Interferon-**γ** (IFN-**γ**)

IFN-γ is named after its ability to interfere with virus growth [24]. It is a pluripotent cytokine that plays a controversial role in the immunomodulation, anti-microbial/anti-viral host defense, allergies [165,166], autoimmune diseases [168], obesity [167], and antitumor immunity [25]. It enhances the response to inflammatory molecules (TLR ligands and TNF [394]), cytotoxic function of NK cells [395], and the number of M1-macrophages [396] to provide phagocytic activity [397,398]. It is an important autocrine signal in the innate immune response and a paracrine signal in the adaptive response [399]. During inflammation, CD4^+^ Th1 cells are the main source of IFN-γ that promotes IFN-γ production by Th1 and NK cells [400,401]. IFN-γ triggers the activation of the proinflammatory response [169] by promoting differentiation of naïve CD4^+^ T cells into Th1 and Th2 cells [171], increasing the killing capacity of CD8^+^ T cells [172] and decreasing the proliferation of Tregs [173]. In DCs, IFN-γ signaling contributes to their maturation, production of IL-12, IL-1β, and activation of CD4^+^ and CD8^+^ T-cells [402]. These cells can be stimulated in an inflammatory or tumor microenvironment by antigens secreted by the tumor or pathogen [403], IFN-γ itself via positive feedback [404], or IL-12, IL-15, and IL-18 [405,406] to activate IFN-γ production. IFN-γ has been shown to interact with a heterodimeric receptor composed of two subunits, IFN receptor 1 (IFNGR1) and IFN receptor 2 (IFNGR2), expressed on the surface of nearly all types of cells to regulate the immune response [407,408]. Secreted IFN-γ binds to its receptor (IFNGR1/2) and can activate the JAK(1/2)-STAT pathway (1/3/4) [160,161], AP-1 [409] and subsequently the up-regulation of interferon regulatory factor 1 (IRF1) and interferon-stimulated genes (ISGs), including those for MHC presentation [394,410,411].

In the TME, the concentration of IFN-γ determines whether the function will be anti or pro-tumorigenic. A high dose of IFN-γ stimulates JAK-STAT1 signaling [155,412,413] and can induce cancer cell death and apoptosis. However, low doses of IFN-γ produced at the tumor site by host-infiltrating cells or during immunotherapy can enhance tumor cell survival, induce risk of metastasis and expression of EMT transcription factors [174,175] via activation of ICAM1-PI3K-Akt-Notch1 signaling in cancer cells [414]. IFN-γ production by NK, NKT, CD8^+^ T, Th1, and γδ T cells stimulates and enhances immunorecognition, recruitment of immune cells to tumor sites and subsequently increases the ability to kill tumor cells [406], by improving the anti-proliferative status of cancer cells (growth inhibition, cell death, autophagy) [415], tumor cell antigenicity, and metastasis reduction by up-regulating fibronectin [155]. IFN-γ arrests the cell cycle and initiates apoptosis in tumor cells (up-regulation of p21 and p27 [416], granzyme B and perforin [417,418]). In addition, IFN-γ can increase the destruction of established tumor-associated blood vessels [419,420], inhibit the migration of TAMs to enhance the efficacy of anti-PD1 antibody therapy [421] (Figure 1, Figure 2 and Figure 3).

Chronic high-dose IFN-γ release [422], loss of the IFN-γ receptor [423], prolonged IFN-γ signaling in tumor cells, and PD-L1 expression (cancer and immune infiltrating cells) [424,425], may induce apoptosis in CD4^+^ T-cells, suppress immune and secondary antitumor immune response [176], cause immune evasion adaptive immune resistance to immune checkpoint therapy [177]. Therefore, IFN-γ is believed to be one of the critical factors determining the success of immunotherapy because dysregulation of IFN-γ responsiveness and/or signaling is often associated with resistance to immunotherapy [426,427,428,429].

## 12. Interleukin 10 (IL-10)

IL-10 is known as human cytokine synthesis inhibitory factor [430], with multiple and pleiotropic effects in immunoregulation, infection, inflammation, autoimmunity, transplantation, and tumorigenesis [431]. It is a cytokine with potent anti-inflammatory properties by repressing the expression of inflammatory cytokines, such as TNF-α, IL-6, and IL-1 and is considered one of the most crucial immunosuppressive cytokines in tumor progression. In general, IL-10 is mainly produced by Th2, Th1, Treg [178], and Th17, and also by CD8^+^ T cells, monocytes, macrophages, DCs [179], B cells [180], mast cells, eosinophils [181], keratinocytes, epithelial cells, and even some tumor cells [182,183]. The IL-10-producing cell type depends on the inducing stimulus, e.g., Th1, Th17 cells, and macrophages represent an important source of IL-10 in infectious diseases [432].

IL-10 receptor is a two-receptor complex consisting of two copies of IL-10 receptor 1 (IL-10R1) and IL-10R2 [430]. Both subunits belong to the class II cytokine receptor family. IL-10R1 forms specific high-affinity interactions with IL-10, whereas IL-10R2 is a shared low-affinity receptor that participates in receptor complexes with other cytokines, such as IL-22, IL-26, IL-28, IFN-γ, and IL-29, which play critical roles in host defense [433]. DCs [184], T cells, B cells, NK, Treg, and mast cells express IL-10R1, whereas the IL-10R2 subunit is ubiquitously expressed. In humans, IL-10 first binds to IL10R1, and then this complex binds to IL10R2, forming a heterotetramer (two IL10R1/two IL10R2), allowing the assembly of the IL-10R complex, which is the first step in the initiation of IL-10 signaling pathways. Once the complex is assembled, Jak1 and Tyk2 associated with IL-10R1 and IL-10R2, respectively, are activated and phosphorylate the intracellular cytoplasmic tails of the IL10R1 subunit. This results in the recruitment and phosphorylation/activation of STAT3 and STAT1 or STAT5 under certain conditions [430]. STAT3 is most notably associated with IL-10 signaling, recruited to IL-10R1 [434] upon IL-10 binding, driving the expression of anti-inflammatory mediators that block various inflammatory pathways. The silencing of STAT3 and suppressor of cytokine signaling 3 (SOCS3) protein reduces the IL-10 expression [435]. In addition, IL-10 is involved in MAPK inhibition and/or activation of a PI3K/AKT inhibitory pathway [183] and can inhibit NF-κB translocation to the nucleus and DNA binding [436], whereas hyper induction and production of high levels of IL-10 are due to MAPK/ERK activation. In macrophages, IL-10 activates the PI3K/Akt/GSK3 β-signaling cascade and modulates downstream transcription [437] and PI3K-mediated mTORC1 activity in monocytes [438]. The IL10 response leads to the expression of anti-inflammatory mediators that block various inflammatory pathways, therefore has prominent role in regulating intestinal inflammation, tumor immunosuppression, viral infection, allergic reactions [185]. For example, IL-10 down-regulates the expression of co-stimulatory molecules on macrophages, differentiation and maturation of DCs, activation of CD4^+^ Tcells [188,189]. It also inhibits the production of proinflammatory cytokines such as IL-1β, IL-6, IL-8, IL-12, IL-18, CSF, and TNF-α, suppresses Th1-associated cytokines (IL-2, IFN-γ) [186], and stimulates B and NK cell survival and proliferation, as well as their production of antibodies and cytokines [187].

The role of IL-10 in modulating the tumor immune response appears to be dependent on the TME and the number of IL-10 receptors expressed on immune cells. IL10 is primarily produced and expressed by M2/TAM, lymphocytes and cancer cells [439]. It contributes positively to tumor growth and promotion (STAT3 in cancer cells), angiogenesis, tumor escape, and metastasis [191]. Furthermore, IL-10 levels increases TGF-β excretion in Treg cells and macrophages and then promotes EMT [190]. Similarly, TGF-β in combination with IL-6, induces IL-10 secretion by Th-17 cells [440]. In addition, IL-10 can suppress T-cell proliferation and activity in breast cancer [191] and inhibit T-cell-stimulated anti-tumor immunity by down-regulating MHC class II (APC). IL-10 produced by TAMs and activation of the IL-10/STAT3/BCL-2 signaling pathway have been reported to contribute to therapeutic resistance in irradiation, chemotherapy and immunotherapy [441,442] (Figure 1 and Figure 2).

Therefore, targeting IL-10 may provide a new therapeutic opportunity for cancer. The use of PEG-IL-10 (pegilodecakin) and anti-PD-1 (pembrolizumab or nivolumab)) treatment to suppress Il-10 signaling [443] or PEGylated human IL-10 (PEG-rhuIL-10), has been designed for clinical use in patients with advanced solid tumors. These strategies are effective in stimulating IFN-γ secretion, perforin, and granzyme B production by CD8^+^ T cells and decreasing TGFβ levels [444].

## 13. Conclusions

The microenvironment significantly affects immune cell response, activation, differentiation, and cytokine secretion. It can enhance the pro- and antitumorigenic response, mediate inflammation and oncogenesis depending on cytokine interference. Many solid tumors, including breast cancer, are composed of heterogeneous cell populations that interact in complex networks through a mixture of cytokines. These are generated by innate or adaptive immune cells, immunocompetent cells, stromal cells, and some cancer cells to inhibit tumor growth, such as IL-2, IL-12, IFNs, or promote tumorigenesis, proliferation, and/or invasion, such as, IL-23, IL-17, TGF- β, IL-10, and some adipokines including leptin, TNFα, IL-1β, and IL-6. Adipokines may contribute to an obesity-related state of low-grade inflammation, regulate energy expenditure, inflammation, cancer growth, progression, and metastasis. Therefore, inflammatory cytokines generated by cancer-associated cells, such as TAMs, CAFs, Th17s, and Tregs, can be expressed by both immune and cancer cells and promote breast cancer, and especially cancer-associated inflammation, such as obesity-associated breast cancer. Thus, they induce tumor-associated inflammation, proliferation, survival, progression, escape, migration, invasiveness, and metastasis of cancer. They decrease the infiltration of Th1, CD8^+^ T cells, NKs, Tregs, promote the infiltration of Th2, M2/TAMs and CAFs and their secretion of pro-tumor immunosuppressive cytokines. Figure 2 summarizes the crosstalk and the immunoregulation between all of these actors. Although they can be activated, such as JAK/STAT, PI3K, AKT, Rac, MAPK, JunB, cFos, and mTORC, involved in the activation of proliferation, survival, differentiation, and cell migration, NF-κB, JNK, and P38 are involved in inflammation, metastatic protein (MMP, COX-2), and anti-apoptotic protein (Bcl-XL). Thus, it forms an immunosuppressive microenvironment to enhance tumorigenicity, cancer cell motility (via MMP9 expression, PI3K-Akt signaling), EMT (FoxC1 in cancer cells), and angiogenesis (activation of Ras/MAPK/PI3K, NF-κB/VEGF). In the Figure 1, TYK2-mediated JAK/STAT signaling network is represented. Moreover, for tumor clearance and effective anti-tumor responses, many cytokines attract effector immune cells, such as NKs, NKTs, M1s, CD8^+^ T cells, APC, and Th1, to regulate immune surveillance, cytotoxic activity, and anti-apoptotic factors for CD8^+^ T cells. In addition, they can stimulate the secretion of perforin, proteolytic enzymes, and cell adhesion molecules and inhibit tumor induced Treg cells that suppress effector T cells and impair the body’s immune response against cancer. Because of the critical role of cytokines in the progression of various disorders, such as cancer, understanding the crosstalk between cytokines can provide important insights into immune-related mechanisms of cancer development, and this knowledge can then be applied to cancer treatment.

## Figures and Tables

**Figure 1 ijms-24-04002-f001:**
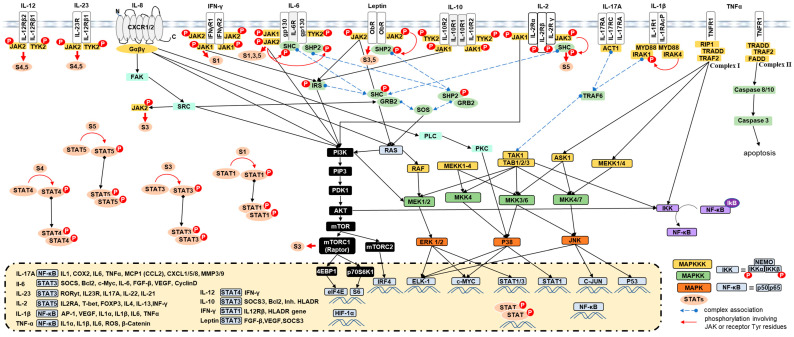
Schematic representation of TYK2-mediated JAK/STAT signaling network. Binding of cytokine to the cytokine receptor which consequently phosphorylates JAK proteins as the cytokine receptor itself lacks intrinsic biological activity. Activated JAKs induce the phosphorylation of STATs which, following dimerization, translocate into the nucleus and stimulate gene expression. JAKs activate other downstream signaling cascades, including PI3K/mTOR, RAS and NF-κB. 4EBP1: eukaryotic translation initiation factor 4E-binding protein 1; ACT1: NF-κB activator 1; AKT: protein kinase B; ASK: apoptosis signal-regulating kinase; ELK: E26 transformation-specific like-1 protein; ERK: extracellular signal-related kinase; FADD: Fas-Associated protein with Death Domain; FAK: focal adhesion kinase; GRB2: growth factor receptor-bound protein 2; HIF-1α: hypoxia-inducible factor 1-alpha; IKK: serine-specific IκB kinase; IRAK: interleukin-1 receptor-associated kinase; IRF4: interferon regulatory factor 4; IRS: insulin receptor substrate; JAK: Janus kinase; JNK: c-Jun N-terminal kinases; MAPK: mitogen-activated protein kinase; MEK/MKK: mitogen-activated protein kinase kinase; MEKK: mitogen-activated protein kinase kinase; mTOR: mammalian target of rapamycin; MYC: myelocytomatosis oncogene; MYD88: myeloid differentiation primary response 88; NF-κb: nuclear factor kappa-light-chain-enhancer of activated B cells; PDK1: phosphoinositide-dependent kinase-1; PI3K: phosphoinositide 3-kinases; PIP3: phosphatidylinositol-3,4,5-trisphosphate; PKC: protein kinase C; PLC: phospholipase C; RAF: rapidly accelerated fibrosarcoma; RAS: rat sarcoma virus; RIP: ribosome-inactivating protein; SHC: SHC (Src homology 2 domain containing) transforming protein; SHP2: Src homology region 2-containing protein tyrosine phosphatase-2; SOS: son of sevenless; SRC: proto-oncogene, non-receptor tyrosine kinase; STAT: signal transducer and activator of transcription; TAB: mitogen-activated protein kinase kinase kinase 7-interacting protein; TAK: mitogen-activated protein kinase kinase kinase 7; TRADD: tumor necrosis factor receptor type 1-associated DEATH domain protein; TRAF: tumor necrosis factor receptor-associated factors; TYK: tyrosine kinase.

**Figure 2 ijms-24-04002-f002:**
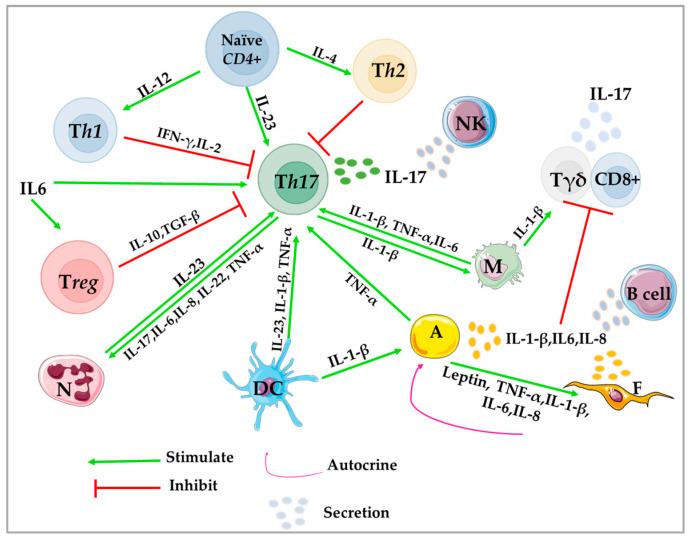
Reciprocal crosstalk and immunoregulation between immune cells, adipocytes, and fibroblasts. DC: dendritic cell; IFN: interferon; IL: interleukin; M: macrophage; N: neutrophil; NK: natural killer cell; Th: T-helper cell; Treg: regulatory T cell; γδT: gamma delta T cells; Th17: T-helper cell 17; A: adipocyte; F: fibroblast.

**Figure 3 ijms-24-04002-f003:**
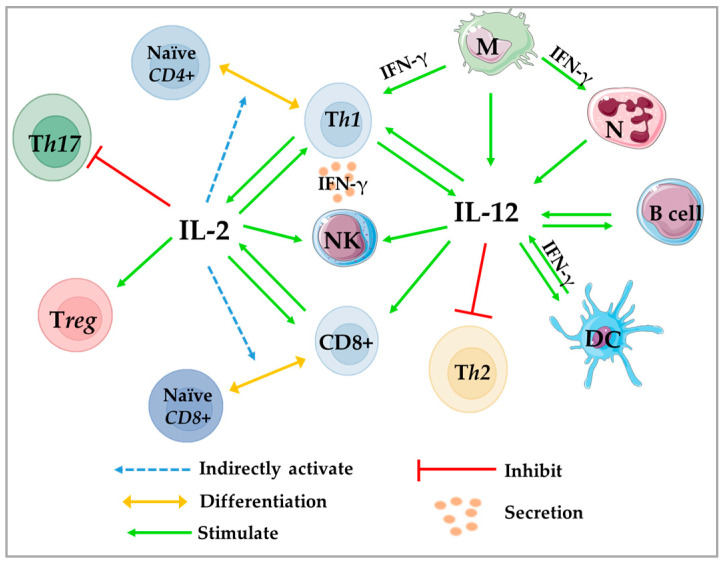
Reciprocal crosstalk and immunoregulation between immune cells by IL-12 and IL-2. DC: dendritic cell; IFN: interferon; IL: interleukin; M: macrophage; N: neutrophil; NK: natural killer cell; Th: T-helper cell; Treg: regulatory T cell; γδT: gamma delta T cells; Th17: T-helper cell 17.

**Table 1 ijms-24-04002-t001:** Main sources, target cells, receptor types, activated pathways, and functions of each cytokine.

	Source	Targets	Action	Receptors	Pathways	Functions
**Leptin**	Adipose cells, enterocytes, CAFs, some cancer cells	Adipose cells, epithelial cancer cells, cancer stem cells, immune cells, endothelial cells, potentially fibroblasts	Endocrine, paracrine, and autocrine	ObR	JAK2/STAT3; MAPK/ERK; PI3K/Akt/Rac [34,35].	Regulates the energy balance, suppressing food intake, controlling appetite and body weight [36,37]; Increases cancer and immune cell proliferation, anti-apoptosis, migration, invasion, angiogenesis [38], EMT [39] and cytokine secretion [40].
**TNFα**	Adipocytes, macrophages, CD8^+^ T, CD4^+^ Th1, NK cells, mast cells, fibroblasts, osteoclasts, endothelial, DCs, Th17, TAMs, epithelial, and malignant cancer cells [41,42,43,44]	Epithelial cancer cells, cancer cells, immune cells, endothelial cells, potentially fibroblasts	Endocrine, paracrine, and autocrine	TNFR1/TNFR2	NF-κB [45,46], JNK, MAPKs, AKT, AP-1, TAZ, JNK/P38 (activate AP-1); Non-canonical NF-κB [47]; MAPK/ERK.	Up-regulates transcription of pro-inflammatory genes, including anti-apoptotic proteins, cell-adhesion molecules, inflammatory cytokines, and chemokines [48,49]; Activates cell survival and proliferation, VEGF production, angiogenesis [50,51], and cell migration; Cancer cell survival and/or proliferation [52], tumor-promoting, aggressiveness [53], EMT, MMP9 expression; IL-17 production by Th17 cells [53].
**IL1-β**	Macrophages, adipocytes, monocyte, DCs, fibroblasts, B-cells, TAMs [54,55], and some cancer cells [54,56]	Cancer cells, Th cells, B cells, NK cells, γδT cells, macrophages, endothelial cells [57].	Paracrine and autocrine	CD121a/IL1R1, CD121b/IL1R2	NF-κB, STAT1, PI3K/Rac [58]; (ERK)1/2, AP-1 [59].	Increases COX-2 expression in cancer cells for cancer progression [60,61] and regulates maturation and proliferation of B cells, activation of NK [62]; Can activate Th, neutrophils to dampen the CD8^+^ T cells [63], γδT and Th17; Induces migration/invasion [64,65], EMT [66], angiogenesis (viaVEGF, neo-angiogenesis, CXCL2) [67,68], and matrix-remodeling activities [69,70].
**IL-6**	Monocytes, macrophages, TAM [71,72], T cells, B cells, fibroblasts, CAFs [73,74], endothelial cells, and adipocytes [75,76,77] some cancer cells [78,79], myeloid-derived suppressor cells (MDSC), and CD4^+^ T cells	Activated B, DCs, cells, T cells, CD4^+^ T, plasma cells, hematopoietic stem cells, cancer cells, macrophages, and endothelial cells [80]	Endocrine, paracrine, and autocrine	IL-6Rα/gp80 IR6Rβ/grp130	JAK1, JAK2, and Tyk2 [81]; STAT3/STAT1 [82,83]; SHP2, ERK, MAPK PI3K [84,85] and mTOR.	Mediates tumorigenesis, increasing cell-cycle progression, resistance to apoptosis and senescence [86], tumor cell proliferation, survival [87,88], and metastasis [89,90]; Blocks DC differentiation, thereby preventing T cell activation and inducing T cell death [91], Th17 cells [92]; Promotes the mobilization of anti-tumor CD8^+^ effector T cell responses, the development of APC, such as DCs and cytotoxic T cells [91,93], as well as the survival, proliferation, differentiation, and recruitment of leukocytes [94,95].
**IL-8/CXCL8**	Macrophages [96], TAMs [97], monocytes [98], fibroblasts [99], epithelial cells [100], vascular endothelial cells [101], CAFs [102], T cells, and some cancer cells	Macrophages, TAMs, monocytes, fibroblasts, endothelial cells, CAFs, T cells, neutrophils [103,104], and some cancer cells [105]	Paracrine and autocrine	CXCR1/IL8RA, CXCR2/IL8RB	PI3K [106], PKB/Akt [106], MAPK [103,104]; Raf-1/MEP/ERK1 cascade, p38 MAPK [107] and PLC; FAK [108,109], STAT3 [97,110], JAK2/STAT3/Snail [96]; Ras/MAPK/PI3K [111,112], NF-κB/VEGF activation [113].	Activates cell survival, angiogenesis, and cell migration [114], cell motility, invasion, and metastasis [115]; Induces M2/TAM macrophage polarization [116] and alters NK cell function [117], EMT [118], and chemoresistance [101], suppresses CD8^+^ T-cell activity [119] and limits the anti-PD-1 immune response. Induces of inflammatory cells recruitment to exert cytotoxic activities [120]
**IL-23**	DCs, phagocytic cells, monocytes, neutrophils, and innate lymphoid cells (ILCs) [121,122,123]	T cells, NK, NKT cells, tumor cells, monocytes, macrophages, and DCs [124]	Paracrine and autocrine	IL23R	TYK2/JAK2/STAT3 [124,125]; NFκB.	Regulates Th17 cell differentiation, stimulates IL-17 production, maintains suppressive Treg activity; Decreases the infiltration of CD4^+^ and CD8^+^ T cells [126]; Enhances tumor-associated inflammation, tumor growth, metastasis [127], angiogenesis [128,129], immunosuppressive cytokines [126]; Increases the expression of VEGF, MMP9, CD31, and the proliferative marker Ki67 in tumors.
**IL-17A**	T helper 17 cells (Th17) [130,131], T-cells [132], CD8^+^ T cells, γδ T cells, and NKT, NK	Epithelial cells, endothelial cells, cancer cells, CD4-CD8- T cells, other T-cells [132], fibroblasts, keratinocytes, and macrophages	Paracrine and autocrine	IL-17R IL17RA/IL17RB	ERK1/2 phosphorylation, p38/MAPK and STAT3; NF-κB.	Amplifies the inflammatory response, the secretion of inflammatory cytokines, including IL-6, TNFα, and IL-1β; Enhances the production of chemokines, such as CXCL-8 which leads to granulocyte recruitment at inflamed sites; Activates oncogenic signal transducer, tumor growth [133], migration [134], tissue invasion, tumorigenesis, inhibits apoptosis, and angiogenesis.
**IL-12**	DCs, B cells, T cells, and macrophages	T cells, NK cells, NKT cells, monocytes, macrophages, DCs, CD4^+^ T cells, and cancer cells	Paracrine and autocrine	IL-12Rβ1IL-12Rβ2	JAK2 and TYK2 [135], STAT4.	Activates M1 macrophage polarization and enhances anti-tumor cytotoxic immune responses in tumor microenvironment [135]; Regulates Th1 cell differentiation and cytokine secretion including IFN-γ; Upregulates MHCI on tumor cells to facilitate antigen presentation; Activates Th1, recruits cytotoxic T cells, NK, and CD8^+^ T cells; Activates the differentiation of naïve CD8^+^ T cells to the effector phenotype and acts as an anti-apoptotic factor.
**IL-2**	Th1-cells, CD4^+^ T, CD8^+^ T cells [136], activated DCs and NK cells [137,138], NK [139], B [140], T [141] cells, neutrophils [142], and some tumor cells [143]	B cells, NK cells, macrophages, CD4^+^ and CD8^+^ T cells, mature DCs, endothelial cells [144,145,146], Tregs, NK cells, and tumor cells	Paracrine and autocrine	IL-2Rα (CD25)IL-2Rβ (CD122)IL-2Rγ (CD132)	JAK1 and JAK3, STAT5A/STAT5B STAT1 and STAT3; PI3K-AKT, JAK-STAT, and MAPK/ERK.	Differentiates CD4^+^ T cells into Th1 and Th2 [147], promotes CD8^+^ T cells, inhibits Th17 differentiation but also expands Th17 cells [148]; Increases NK cytolytic activity, mediates activation-induced cell death, induces development of Treg (FoxP3) and cytotoxic T cells (CD8^+^ T cells); Activates [149] tumor cell growth, survival, and differentiation, cytokine production (Il-4, Il-12,Il-6) [150,151], and activates induced cell death in diverse immune cell types [152,153].
**IFN-γ**	NK and NKT in innate immunity, macrophages, epithelial cells, Th1 [154], DCs [155], Tγδ [156], and CD8^+^ T cells [157,158] in the adaptive immune response [159]	T-cells and NK, cancer cells, macrophages, Treg, endothelial cells, Tγδ [156], CD4^+^ T, and CD8^+^ T cells [157,158]	Endocrine, paracrine, and autocrine	IFNGR1/2	JAK(1/2)-STAT(1/3/4) [160,161]; JAK-STAT; MAP, PI3K, JNK, and NF-κB [162,163,164]; Src kinases/MAPKs/ERK/p38/then Fos and Jun kinases; ICAM1-PI3K-Akt-Notch1.	Implicated in allergies [165,166], obesity [167], autoimmune diseases [168], and cancer; Activates the proinflammatory response [169] by promoting NK cell activity [170], differentiation of naïve CD4^+^ T-cells into Th1 and Th2 cells [171], increases the killing capacity of CD8^+^ T-cells [172] and decreases the proliferation of Tregs [173]; Eliminates tumors [154], reduces metastasis by up-regulating fibronectin [155], arrests the cell cycle and initiates apoptosis in tumor cells, inhibits the migration of TAMs; May enhance tumor cell survival, induces risk of metastasis, EMT transcription factors [174,175]; May induce apoptosis in CD4^+^ T-cells, suppress immune and secondary antitumor immune response [176], causes immune evasion adaptive immune resistance to immune checkpoint therapy [177].
**IL-10**	Th2, Th1, Treg [178], Th17, and also by CD8^+^ T cells, monocytes, macrophages, DCs [179], B cells [180], mast cells, eosinophils [181], keratinocytes, epithelial cells, and even some tumor cells [182,183]	DCs [184], T, B, NK, Treg, mast, dendritic cells, M2/TAM lymphocytes, and cancer cells458	Endocrine, paracrine, and autocrine	Two IL-10R1 and two IL-10R2	Jak1 and Tyk2; STAT3/STAT1/STAT5; MAPK inhibition and/or activation of a PI3K/AKT inhibitory pathway.	Leads to the expression of anti-inflammatory mediators that block various inflammatory pathways; Regulating intestinal inflammation, tumor immunosuppression, viral infection, allergic reactions [185]; Inhibits the production of proinflammatory cytokines, such as IL-1β, IL-6, IL-8, IL-12, IL-18, CSF, and TNF-α) [186], suppresses Th1-associated cytokines (IL-2, IFN-γ, and stimulates B cell and NK cell survival and proliferation, as well as their production of antibodies and cytokines [187]; Downregulates the expression of co-stimulatory molecules on macrophages, differentiation, and maturation of DCs, activation of CD4^+^ T cells [188,189]; Inhibits NF-κB translocation; Contributes to tumor growth and promotion (STAT3 in cancer cells), angiogenesis, and metastasis. In carcinomas, IL-10 levels increase TGF-β excretion in Treg cells and macrophages and then promote EMT [190]; Suppresses T-cell proliferation and activity in breast cancer [191] and inhibits T-cell-stimulated anti-tumor immunity by down-regulating MHC class II (APC) and class I (colon tumor cells) [192].

## Data Availability

Not applicable.

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
