# Peer review of "Crosstalk of Inflammatory Cytokines within the Breast Tumor Microenvironment"

_ijms, 2023, doi:10.3390/ijms24044002_

Round 1

Reviewer 1 Report

The article reads well and has referenced most recent publications

Author Response

As suggested by reviewer 2, we have refocused the manuscript on the breast tumour microenvironment. We have changed the title accordingly: « Crosstalk of inflammatory cytokines within the breast tumor microenvironment ».

The introduction was not changed: we want to keep a general introduction on cytokines. The focus on breast cancer is made in the last paragraph of the introduction.

The entire text describing the activities of each cytokine has been completely revised in order to provide data exclusively in the context of breast cancer. In addition, deletion of data that may be redundant has been made.

The paragraph at the end of each section, which explains how targeting each cytokine might be therapeutically useful, has been rewritten to make the text clearer and more concise and to focus on the breast cancer context.

The figures have been completely revised to allow a better understanding of the interactions we have described in the text.

Reviewer 2 Report

This is a very thorough review of the literature on the crosstalk of inflammatory cytokines in the tumour microenvironment.  However, the authors may have been overly ambitious in their review as became increasingly clear in reading this manuscript.  The authors state that they are focusing primarily on breast cancer, and it may be better if they more tightly focussed on this disease as the complexity of review made it very difficult reading and in the end I gained few new insights. Part of the problem may be that different tumour types will respond differently to individual cytokines, so limiting the focus to a single or closely related tumour types, e.g. breast and ovarian cancer would help to reduce the complexity of the information presented.  A problem is the very thorough nature of the review.  In the end, it clear that authors have really combed the literature, citing widely, but as it currently stands the manuscript does little more than compile the literature rather than a condensation of the information in those cited papers and relating the information to the topic in a clear and concise manner that assists the reader in understanding the topic, in this case cytokine crosstalk.  The structure of the review is fine, separating into individual cytokines.  But in each section, it was difficult to understand how the crosstalk was regulated and what it was regulating.  Two of the sections towards the end had diagrams to assist the reader, although by then I was so confused that they did not assist as much as they should.  There were many sections where concepts were re-introduced on several occasions which made the section unnecessarily longer and less clear. I have highlighted a couple of examples below.  I did appreciated the paragraph towards the end of each section that discussed how targeting each cytokine might be useful therapeutically, but in most cases no single or simple recommendation could be provided in part due to the different effects of cytokines in the different cancer types.  I recommend that the authors go back over each individual cytokine section and condense the information into a more accessible form.  Diagrams would assist with this.  The level of detail was in many cases unnecessary, for example, the cellular signalling pathways that each cytokine activates need only be introduced once, it is more important to tell the reader which cell types this was affecting, and importantly how this might influence tumour growth.  This information was presented, but again in a form that did not provide the reader with a clear or easily accessible understanding.   The major focus of the review is crosstalk, and again this was presented in each section but in a manner that did not assist the reader in gaining a clearer understanding.  As this is the primary aim of a review such as this, more work needs to be done to achieve this goal. 

Specific points:

The first more than 2 pages define many of the terms used throughout the manuscript, yet these are redefined multiple time after this.  This should be rectified.

Table 1 identifies the signalling pathways activated by each cytokine.  This should be sufficient for this aspect of the review and can be easily referred to in the text rather than re-introduced for each cytokine, multiple times.

Line 183; this is all very unclear, not sure what is meant here?  Is this about blocking Abs of Ob-R?

Line 223; endothelial/epithelial tyrosine kinase (ERK)?

Line 314; IL-6 activation of STAT3 is introduced twice in this section.  There are several other examples of this with other features introduced multiple times within a section. Please check this and modify the text to make this easier for the reader to follow.   

Line 351; “limit the anti-PD-1 immune response to cancer” Not clear what is meant by this.

Line 373; TME?

Line 374-380; so does this approach work?

Line 404; this is unclear. What is IL-6/TGFb, and IL-23 promotes Th17 expressing IL-17 as stated above.  Please simplify this section as again there is the re-introduction of previously introduced concepts within individual sections. 

Line 411; How does IL-17A enrich Tregs?  Is it a chemokine for Tregs or does it drive the differentiation of CD4 to Tregs?  And what is consequence of Treg induction of IL-17B expression in cancer cells?

Line 460-463; this was introduced and discussed extensively int he previous section.  A very brief statement or refer to above section would make this easier to read. 

Line 471-472; this is unclear

Line 489-490; Are these inhibitors of IL-23 processing?

Reviewer 3 Report

In this well-written manuscript, the authors made a comprehensive summary regarding the crosstalk of inflammatory cytokines within the tumor microenvironment.

They focused on the cytokine pathways involved in cancer immune responses and their applications in anti-cancer therapeutics.

Overall, this paper is well organized and provide a great reference for the researchers in this area. I suggest the publication of this paper after minor revisions.

It is better to modify the figures, as the current version seems to be rough.

Author Response

(The authors gave the same response as above.)

Round 2

Reviewer 2 Report

The have responded adequately to the comments in my initial review.  This has improved the focus and clarity of the review.